# Accidental Landmarks: How Showing (and Removing) Emphasis in a 2D Visualization Affected Retrieval and Revisitation

Aristides Mairena*
University of Saskatchewan

Md. Sami Uddin†
McGill University

Carl Gutwin‡
University of Saskatchewan

## ABSTRACT

Many visualizations display large datasets in which it can be difficult for users to find (and re-find) specific items. In systems that provide highlighting tools (e.g., filtering or brushing), emphasized points can become "accidental landmarks" – visual anchors that help users remember locations that are near the emphasized points. Accidental landmarks could be useful (by aiding revisitation), but if users become dependent on them, removing or changing the highlighting could cause problems. We provide designers with information about these issues through two crowdsourced studies in which people learned a set of item locations (in visualizations with or without emphasized points); we then removed or changed the highlighting to see if performance suffered. In the first study, which used a simple grid of points, results showed that changing or removing emphasized points significantly impeded users' ability to re-find targets, but the highlighting did not improve performance during training. In the second study, which used a more complex scatterplot, we found that highlighting significantly improved performance during training, but that removing or changing the emphasis points only reduced re-finding performance for a few target types. Our work demonstrates that visualization designers need to consider how transient visual effects such as emphasis can affect spatial learning and revisitation, and provides new knowledge about how visual features can affect performance.

**Index Terms:** Human-centered computing—Visualization—Visualization techniques—; Human-centered computing—Visualization—Visualization design and evaluation methods

## 1 INTRODUCTION

A ubiquitous task in large visualizations is finding and re-finding specific items – to inspect values during exploration or compare results to look for insights [28]. Finding and re-finding can be difficult, however, when objects in visualizations are visually undifferentiated (e.g., dots in a scatterplot), and names or labels are only available through inspection (e.g., hovering over a dot); in many visualizations, finding items for the first time can involve laborious visual search. Once the user finds an item, the problem changes to one of *revisitation* – i.e., finding items that have already been visited. Revisitation can be much faster than visual search if the user can remember where the item was [39, 57]; however, the undifferentiated nature of data items in many visualizations provides little support for users' spatial memory.

One way to support the development of spatial memory – and thus support revisitation – is to include *landmarks* in the visual presentation. Landmarks are obvious visual features that are noticeably different from their surroundings, and that can provide a frame of reference in which users can remember nearby locations based on

---

*e-mail: aristides.mairena@usask.ca
†e-mail: sami.uddin@mcgill.ca
‡e-mail: gutwin@cs.usask.ca

their relative position to the landmark. Structural elements such as corners can be strong landmarks [55], and previous research has also shown that adding artificial landmarks such as coloured blocks can provide valuable anchors for spatial learning when there are a large number of items in the dataset [56].

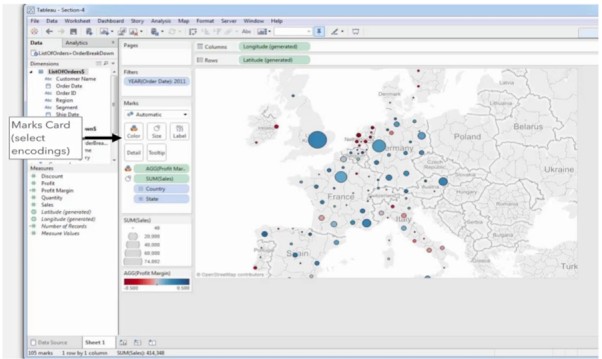

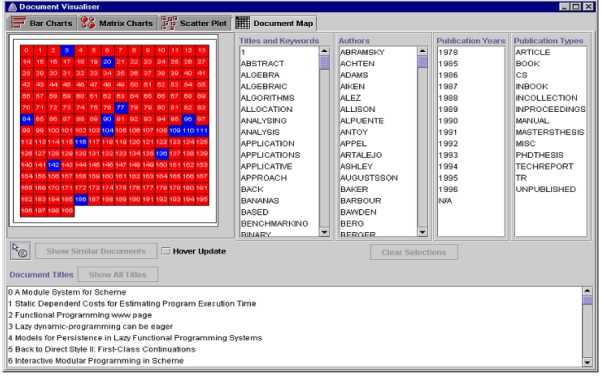

Figure 1: Top: Screen capture of the Tableau visualization tool. Users highlight data points through the "Marks Card" that allows specification of highlights and colours during exploratory data analysis. Bottom: Screen capture of a document explorer tool, highlighting document positions based on filters (from `https://bit.ly/3BQZue2`).

Information visualizations often add visual features such as colour to a set of items in the presentation (through actions such as highlighting a subset of the dataset) and can contain clusters of data that serve as spatial landmarks – but the reason for these features is almost never to add landmarks. Instead, visual highlighting is typically the result of a user operation such as filtering or brushing: for example, the user might set a filter threshold of a third variable to emphasize datapoints in a scatterplot that are above that threshold (see Figure 1) or utilize dynamic queries [61] to hide/show items. We note that in some visualizations all datapoints are coloured or augmented based on an attribute variable, but here we consider representations that only provide a standard glyph for each datapoint.

When a visualization has a subset of datapoints that are visually emphasized, the highlighted points can become "accidental landmarks" – items that have the visual characteristics of landmarks, even though this is unintended by the designer. When users find

and re-find items in a visualization that has some items highlighted, they may start to use the accidental landmarks as anchors for finding nearby items (e.g., "the item I need to remember in the scatterplot is just below the red item").

These accidental landmarks can be useful by providing anchors for revisitation, but they could also cause problems if users become dependent on them, because the highlights could disappear or change (e.g., when a user selects a different subset to emphasize). If a user comes to rely on the visual landmarks, when they eventually need to revisit data points without this aid, they will have difficulty because the aid is missing or different. This phenomenon of users becoming dependent on external aid or feedback is known as the guidance hypothesis [44, 47], which suggests that a reduction in effort provided by guidance during training will lead to poorer retention [17]. However, contrasting research to the guidance hypothesis suggests that spatial knowledge can also be gained through incidental learning [3, 26], which occurs by simply interacting with an environment in a spatial fashion.

These competing hypotheses mean that it is difficult to predict what will happen to spatial learning and revisitation when accidental landmarks occur in visualizations. To determine both the potential benefits and risks of visual emphasis that could be used as landmarks, we carried out two between-participants crowdsourced studies (N=180) to test the effects of highlighting points in scatterplots, and then removing or changing the emphasized items.

In our first study, we asked participants to find and re-find several targets in a simple grid visualization that did not provide strong structural or layout landmarks (other than corners and edges). We tested three conditions: a baseline version with no emphasis, a version with emphasis that was removed after training, and a version with emphasis that was changed to a different subset after training. We measured people's performance in three training blocks where any emphasis effects were present, and in a fourth block where the emphasis was removed or changed.

Results of the first study showed that accidental landmarks did not improve search time or number of hovers during the three training blocks, but did have an effect on performance when removed or changed – in the fourth block, both search time and hovers increased substantially when compared to the no-landmarks condition. In addition, the results were stronger for some targets (e.g., for the target that was emphasized during training, there was a larger detriment to removing / changing the highlighting). Subjective results showed that participants felt that finding targets was more difficult when the highlights were removed or changed.

Our second study tested the same experimental conditions, but in a more-complex scatterplot based on a real-world Gapminder dataset [43]; this visualization had substantially more internal structure that provided additional landmarks (such as clusters of points, edges, and areas of white space). Results of the second study showed that search time and hovers during the learning blocks were both lower with the accidental-landmarks conditions, but there was no significant decrease in performance when the highlighting was changed or removed. We attribute the change in results seen with Study 2 to the additional structural landmarks that were available in the more-complex scatterplot.

Our two studies provide new understanding of how 'accidental' visual features affect visual search, spatial learning, and revisitation in information visualizations. Our findings suggest that in visualizations without extensive structural or layout-based landmarks, participants may become overly dependent on visual emphasis that arises from filtering or brushing. In more complex visualizations, the value of accidental landmarks increases during early use, but the additional landmarks provided by structure and layout appear to mitigate any over-reliance on the highlighting. Our work makes four main contributions. First, we identify a phenomenon – emphasis that provides accidental landmarks in visualization – that has not been considered previously. Second, we provide empirical evidence that emphasis-based landmarks can provide a benefit for visual search (depending on the visualization), but can also cause problems when they are taken away or changed. Third, we provide new knowledge that can guide designers' choices about what emphasis and potential aids to use to support spatial awareness. Finally, we provide possible design improvements for emphasis effects that address some of the issues seen in our study.

## 2 RELATED WORK

### 2.1 Learning and Retrieval

A wide variety of research has been carried out to investigate how humans acquire knowledge and skills. Prior work in psychology has extensively studied human memory [5, 6, 13, 15], how the skills necessary for learning and retrieval are developed [2, 41], the development of learning abilities in children [24], and how sex differences may affect navigation and spatial orientation [35].

Anderson [2] and Fitts *et al.* [18] suggest that skill development occurs in three main stages: cognitive, associative and autonomous. When applied to 2D visual displays, users in the cognitive phase learn items through slow visual search and visual inspection (e.g., finding icons in a toolbar or files in a file browser [17]). In the associative stage, users understand the general contents of the dataset and begin to remember items and locations, allowing faster revisitation for some items. In this stage, however, users still typically perform visual search within a local area after reaching the vicinity of an object of interest. Finally, users in the autonomous stage have memorized item locations, and can recall and revisit an object's location without needing any visual search.

People learn object locations in 2D visualizations as a side effect of interacting with them, and the rate at which locations are learned follows a power law of practice [10]. In previous HCI research, several interfaces have shown the utility of spatial memory to improve performance. For example, Robertson *et al.'s* initial *Data Mountain* study and a subsequent study by Jansen *et al.* which evaluated the approach in a wall display show how the spatial arrangements of thumbnails in a spatial environment allows faster retrieval times than standard bookmarking systems [30, 42]. Similar benefits have also been found in tasks such as list revisitation [21] and command selection in interfaces [54, 58].

### 2.2 Supporting Spatial Learning

Knowledge of the location of an item (be it in a natural environment or digital space) is often relative to other objects or items. People learn, organize, and communicate spatial knowledge by reorganizing the spatial relations among items in an environment [40]. Mou *et al.* suggested that human memory systems use frames of reference to specify the remembered locations of objects [40]: for example, Scarr *et al.* stated that "explicit rectangular boundaries, such as the walls of a room or the edges of a table, can generate a frame of reference" and added that a grid-based item layout can also support spatial knowledge by creating an implicit axis of reference [12].

Previous work on supporting spatial learning has considered two main strategies: spatially stable layouts, and landmarks. Researchers have demonstrated the benefits of laying out interfaces in a way that are spatially stable [46, 56], for example, Gutwin *et al.* and later work by Cockburn *et al.* showed that a stable layout of commands in an interface can improve recall efficiency compared to hierarchical ribbons or menus [11, 21]. Similarly, Scarr *et al.'s* CommandMap showed that spatially stable icon design on a desktop interface improved the recall of icons [45, 46]. The benefits of spatial stability have also been shown in other interfaces such as smartphones [65], tablets [20], smartwatches [33], and virtual environments [19].

Landmarks are a second strategy for improving navigation performance. Landmarks are easily identifiable objects that have distinct spatial features (such as shape, colour, or semantic value [53]) that

can provide a frame of reference for nearby objects. Similar to the benefits of landmarks in real life (e.g., using a prominent building when navigating a city), landmarks have exhibited potential in digital workspaces. Several types of landmark have been considered, such as the corners of a screen or the bezel on a device [20, 48], which can provide a strong reference for nearby objects. However, since these landmarks may not naturally occur in larger workspaces (e.g., there are no corners or edges in the middle of a display), researchers have also examined the use of hands [58] and the idea of adding artificial landmarks (e.g., a background picture, or simple coloured shapes) [56] to assist users in remembering the locations of objects in the visual field.

### 2.3 Emphasis and Attention in Infovis

The goal of emphasis is to manipulate the visual features of a chosen data element to make it visually prominent so that a viewer's bottom-up attention is directed to an element of interest [23]. Many theories have been developed over time to explain how emphasis can guide a viewer's attention. For example, similarity theory developed by Duncan and Humphreys shows that the efficacy of emphasis decreases with increased target/non-target similarity and with decreased similarity between the non-targets [16]. Similarly, the Guided Search theory by Wolfe follows a two-stage process for attention, first guided by visual salience (bottom-up attention) but adding that attention can be biased toward targets of interests (e.g., a user looking for a red circle) by encoding items of user interest: for example, assigning a higher weight to the items with red colour [63].

Another theory, the relational account of attention theory, also follows the premise that if users are given a specific task or have a feature they are interested in (e.g., a user searching for a red circle), attention will be guided to the mark that differs in the given direction from the other marks (e.g., attention will be guided to the reddest circle among all circles displayed) [16, 60].

Similarly, a recently proposed model suggests three main processes for how attention is guided when viewing a visualization: current goals, selection history and physical salience (bottom-up attention) [4]. This model suggests that there is an inherent bias to prioritize items that have been previously selected, which may differ from current goals, and as such, selection history, goal-driven selection and visual salience are competing processes, affecting the effectiveness of emphasis to serve as landmarks.

Consistency is a fundamental guideline in HCI for supporting spatial awareness and memory/recall capabilities [17, 45, 46]. Landmarks are known to supplement the capabilities of an interface by providing anchors that people can use to build better spatial awareness. When interfaces are not visually consistent — such as an interactive visualization which changes depending on actions such as filtering or highlighting — landmarks can provide a method for spatial learning within this uncertainty. However, landmarks in visualization remain relatively unexplored, with questions such as whether removing a landmark (such as when a user removes a highlighting feature in a visualization) or changing the landmarks (e.g., users selecting a different set of objects to highlight) affects the spatial memory of previously learned objects. In addition, there are other factors such as the visual salience of these landmarks, current tasks, and previous selections that may affect how users perform revisitation tasks in a visualization. In the following studies, we set out to determine the effects of using emphasis as a landmark in visualization for spatial awareness and test the limits of emphasis by re-creating common tasks such as removing and changing emphasized objects.

## 3 STUDY 1: EFFECTS OF ACCIDENTAL LANDMARKS IN A SIMPLE GRID VISUALIZATION

We conducted an online experiment to explore whether accidental landmarks in a simple grid visualization would affect spatial location learning and performance, both when the assistance was present and after it was removed or changed. The study asked participants to repeatedly find a set of seven targets in an 8x8 grid that had few structural or layout-based landmarks, other than corners and edges; we recorded search time, hovers required to find a target, and errors.

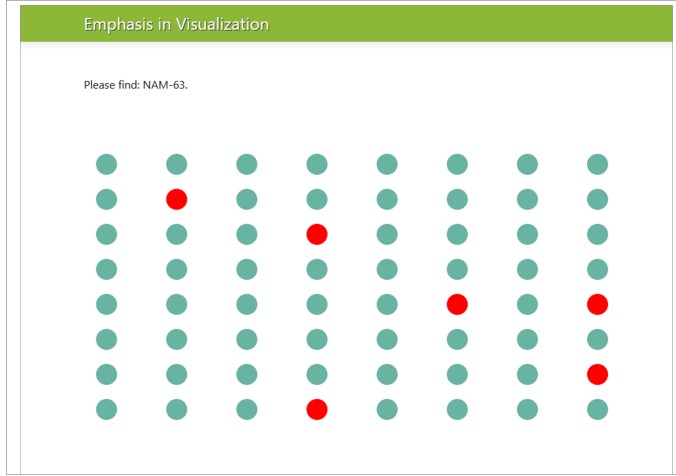

Figure 2: Example of the study system interface that participants would see when completing a trial. In the *No-Landmarks* condition, there were no red highlighted circles.

### 3.1 S1 Study System

A web-based application was developed using HTML, CSS, and JavaScript (D3.js [7]) to display an 8x8 grid of circles that contained targets and distractors (some of which were also accidental landmarks). The interface presented the name of the target, and the user had to click on the target item to confirm a selection. Item names are not permanently visible, but could be shown in a tooltip by hovering the mouse on any item (names were taken from an existing plant-breeding dataset). Hover feedback was immediate (similar to commercial visualization systems), however, we only considered hovers with duration of 300ms for analysis to remove hovers that were simply due to traversing over the items. An example of the study interface is shown in Figure 2.

To ensure that the way we tested targets and emphasis was fairly compared for each task type, we used a grid-style visualization. We used a simple grid for our first study in order to control the number of structural or layout-based landmarks in the visual presentation, and to control the distance between targets and landmarks. Although this style of visualization is less common than other types such as scatterplots, there are still many examples of grid-based visual layout: for example, a visualization of a plant-breeding field trial would typically use a grid to match the arrangement of the physical layout of plots in the field; similarly, the document map shown in 1 organizes items into rows and columns. We also used the same combination of targets and accidental landmarks for all study conditions to ensure equal difficulty. This required that we use a between-participants design for the study.

The study had three conditions that differed in terms of how accidental landmarks were used:

- *No Landmarks:* this condition provided no accidental landmarks – participants saw the plain grid of items, with no red highlights.

- *Landmarks-Removed:* in this condition, participants saw the same grid of items, but with six items coloured red (simulating

a previous filtering operation that had highlighted these items as accidental landmarks). The red highlights were removed in the final block.

- *Landmarks-Changing:* this condition provided the same grid and red highlights as above during the training blocks, but in the fourth block the highlights were moved to a different set of items (rather than being removed altogether).

### 3.1.1  S1 Targets

For the study, seven of the 64 items were used as targets, and six of the 64 were coloured red as accidental landmarks. One of the items was both a target and a landmark. Target positions were sampled from three areas of the grid [56]: three from the corner regions, two from the edges and two from the centre region. Targets and their locations are shown in Figure 3.

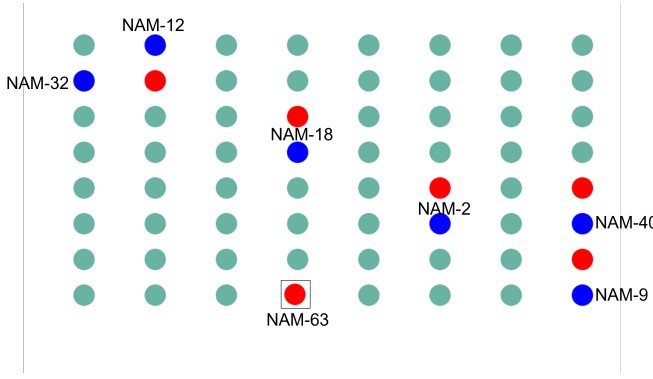

Figure 3: Locations of the targets (shown here in blue, not shown in the study) in relation to the landmarks. Target NAM-63 (shown with a blue square) was both a target and a landmark. In the *No-Landmarks* condition, there were no red highlights.

### 3.2  S1 Procedure

Each condition in the experiment followed seven phases: (1) informed consent, (2) demographics questionnaire, (3) vision test, (4) guided tour, (5) study tasks, (6) post-study questionnaires, and (7) debriefing. The specific questions and tasks for each condition are described in each condition's section below. Participants first completed informed consent and demographics forms, and were then asked to complete an Ishihara test and questionnaire to screen for colour vision deficiencies [29]. Participants then completed a guided tour through all the targets, after which they could proceed to the study.

### 3.2.1  Guided Tour

Participants were first randomly assigned to one of the three study conditions. In the guided tour phase, the experimental system showed the grid (including red highlights if the condition included them). The system then took the participants on a "guided tour" of the seven targets, with each target shown one at a time, highlighted in blue. Participants had to click on the target to proceed to the next target. After all targets were presented, the interface automatically proceeded to the study.

### 3.2.2  Study Phase

After the guided tour, participants completed the study trials. Every trial began by displaying the name of a target at the top of the screen (the name remained visible for the duration of the trial), and participants were asked to find and select the corresponding target item from the grid. Targets were presented in random order (sampling

without replacement); locations of targets (and landmarks if shown) were the same in all conditions. Participants could see item names immediately upon hovering over the item with the mouse. After each correct selection, the screen was blanked for 0.5s to prevent contrast effects between trials. The study consisted of three training blocks in which landmarks (if part of the condition) were shown and a fourth block in which any landmarks were either removed or changed. Because the *Landmarks-Changing* and *Landmarks-Removed* conditions used the same landmarks, this means that these conditions were identical for the first three blocks. In the *No-Landmarks* condition, no landmarks were shown at any point. After completing all blocks, participants were asked to fill out post-study questionnaires, were shown debriefing information, and were compensated for their participation.

### 3.3  S1 Participant Recruitment

We recruited 90 participants ($\mu_{age}$ = 33.15, $\sigma_{age}$ = 10.84, 55 men, 33 women, 2 non-binary) across the three conditions (30 per condition) using Amazon's Mechanical Turk (MTurk), and gathered data through a custom browser-based experiment tool [31]. MTurk is an online platform where requesters can post tasks that workers can opt-in to complete. Data collected from MTurk has been previously used in a variety of human-computer interaction studies [14, 32, 34, 51] and to model perception in visualization [27, 52], including assessing separability of variables [50], measuring colormaps [37], and effectively detecting motion [59]. Using MTurk, however, requires that special care must be taken to ensure the integrity of the data, as bots or negligent workers must be filtered out. Our study required workers to have over 90% HIT acceptance rate (i.e., a measure of the quality of a worker's previous tasks). We also checked the questionnaire responses to ensure that the same answer was not used for all of the questions, as well as whether the study was completed too quickly or too slowly.

All participants were paid $3 for completing the study, which took approximately 15 minutes. Self-reported estimates of monthly visualization usage among participants averaged 33 hours (SD = 66.14) with pie charts, line charts, bar graphs and maps/weather charts as the most commonly used or viewed charts.

### 3.4  S1 Study Design

Our goal was to understand the effects of landmarks on spatial learning and revisitation in visualizations. Our main research questions (RQ) for this study were:

- RQ-1: Do accidental landmarks improve finding and re-finding when they are present (i.e., decreased search time, number of hovers, and error rate)?

- RQ-2: Does removing or changing landmarks after a learning period affect re-finding (i.e., increased search time, hover counts, and error rate)?

To investigate these questions, the study used a mixed factorial design with three factors:

- Condition (between-subjects): *No-Landmarks*, *Landmarks-Removed*, *Landmarks-Changing*

- Target Locations (within-subjects): seven target locations (see Figure 3)

- Blocks (within-subjects): 1-4 (blocks 1-3 are training; block 4 removes/changes any landmarks).

Our primary dependent variables were search time, hover counts (only included if longer than 300ms), error counts (i.e., incorrect clicks), and subjective ratings of difficulty and effort from post-session questionnaires. Targets were the same for all participants.

## 4 S1 STUDY RESULTS

We report effect sizes for significant ANOVA results as generalized eta-squared $\eta^2$ (considering .01 small, .06 medium, and >.14 large [36]). Outliers were determined as any trial with a search time greater than 3 SDs above the block's mean. 73 of the 2520 trials were removed from the analysis. All pairwise t-tests were corrected using the Holm-Bonferroni method.

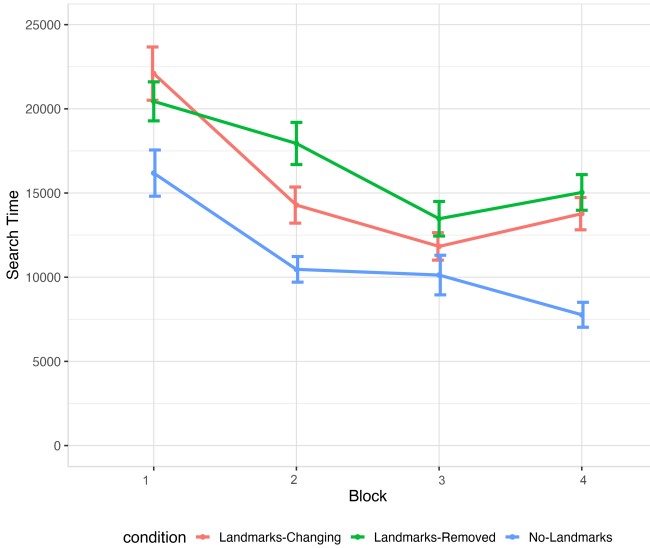

Figure 4: Mean trial search times (± s.e.) across learning blocks (1-3). Block 4 shows the results of removing or changing emphasized objects.

### 4.1 S1 Effects of Landmarks on Learning: Search time, Hovers, and Errors in Learning Blocks

#### 4.1.1 S1 Learning Blocks – Search Time:

Search time in the test trials was measured from the time a target name appeared on the screen to the time the system registered a correct item selection. Search times across all blocks for the three conditions are shown in Fig 4.

For the learning blocks, a 3x3x7 RM-ANOVA (*Condition* x *Block* x *Target*) showed a main effect of *Condition* ($F_{2,178} = 15.21, p < 0.001, \eta^2 = 0.02$), *Block* ($F_{2,174} = 18.00, p < 0.001, \eta^2 = 0.02$) and *Target* ($F_{6,522} = 6.16, p < 0.001, \eta^2 = 0.01$) and an interaction between *Condition* x *Block* ($F_{4,174} = 2.44, p = 0.004, \eta^2 = 0.01$) on search time. Post-hoc pairwise t-tests showed significant differences between *No Landmarks* and both landmarks conditions (both $p < 0.05$).

Across all magnitudes and targets, search time was lowest with the *No-Landmarks* condition (mean 12268ms); the average mean of the landmark conditions was 16685ms (note that both landmark conditions were identical in the learning phase, so any difference between them is due to group differences). To further investigate the *Condition* x *Block* interaction and consider the rate at which the different groups improved, Fig 5 shows a version of the data that normalizes the other blocks based on block 1 performance. Fig 5 suggests that there were group differences in the two landmark conditions, but also indicates that participants in both landmark conditions learned less quickly than the *No-Landmarks* condition.

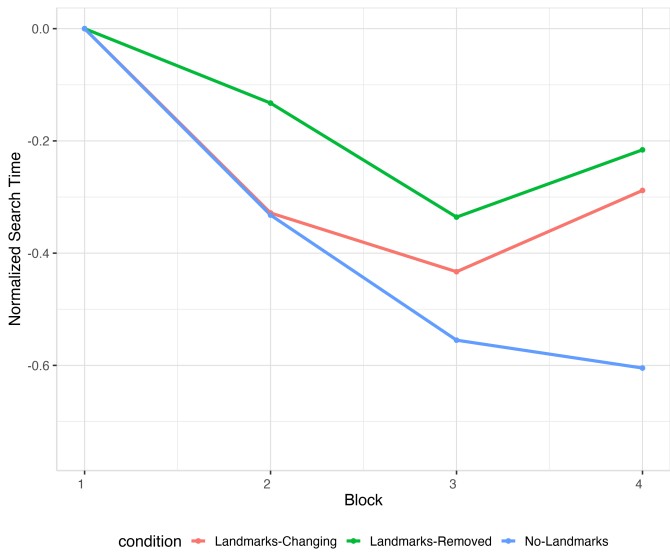

Figure 5: Search time results after block 0 normalization

#### 4.1.2 S1 Learning Blocks – Hovers:

We measured the number of hovers as the number of times the participant held the cursor over a target for 300ms or more to show the name. Hovers are a more sensitive measure of progress through the stages of learning and performance: as a participant moves through the different blocks, there should be a reduction in the number of items that they need to inspect. Mean hovers per trial are shown in Figure 6.

For the learning blocks, a similar 3x3x7 RM-ANOVA (*Condition* x *Block* x *Target*) showed a main effect of *Block* ($F_{2,178} = 11.047, p < 0.001, \eta^2 = 0.03$) and *Target* ($F_{6,522} = 2334, p < 0.001, \eta^2 = 0.02$) on hover count, and also showed two interactions: *Condition* x *Block* ($F_{4,356} = 1.84, p < 0.01, \eta^2 = 0.01$) and *Block* x *Target* ($F_{12,1044} = 1.84, p < 0.001, \eta^2 = 0.01$). However, the ANOVA found no main effect of condition (p=0.37). Across all magnitudes and targets, hovers were similar in all conditions, with the lowest mean in the *No-Landmarks* condition (10.12 hovers per correct selection) compared to the averaged mean of the landmark conditions at 10.81 hovers. We again investigated the *Condition* x *Block* interaction to consider learning rates. Fig 7 represents the same data, taking block 1 as a baseline and normalizing the following blocks based on block 1 performance. Fig 7 again suggests group differences in the landmark conditions, and shows similar learning rates across all three conditions.

#### 4.1.3 S1 Learning Blocks – Errors:

We measured errors as the number of incorrect clicks before choosing a correct target. As participants could hover over targets until they found the correct one, errors were low overall, with an average of 0.63 errors per trial across all conditions and blocks. For the learning blocks, A 3x3x7 RM-ANOVA (*Condition* x *Block* x *Target*) showed no main effect of *Condition* (p=0.42), *Block* (p=0.08), or *Target* (p=0.59) on errors.

#### 4.1.4 S1 Learning Blocks – Target-by-Target Analysis:

As the ANOVA results showed a main effect of *Target* on search time and hover counts, we looked into the results for each specific target. Overall, as seen in Figures 8 and 9 we found that the targets that required the fewest hover actions and were found fastest were

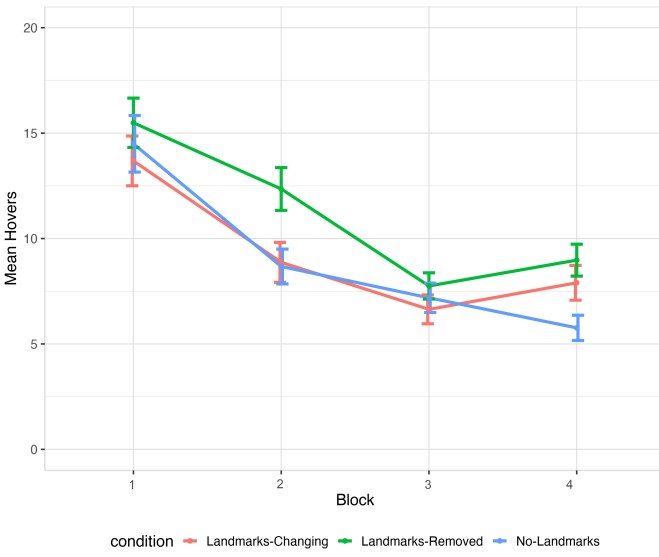

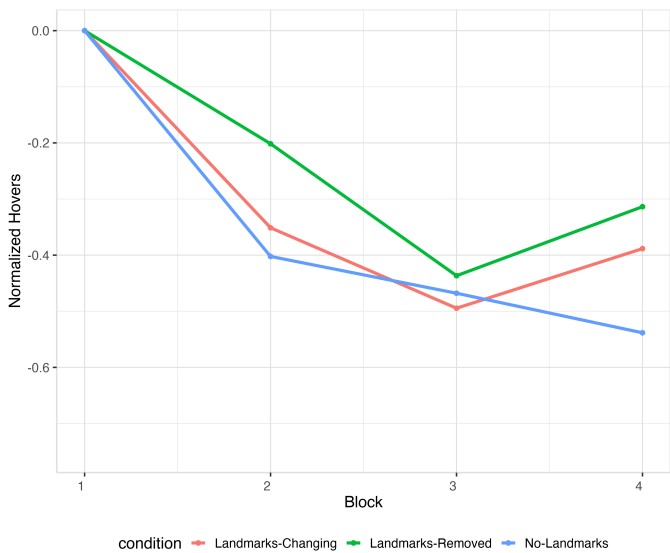

Figure 6: Mean trial hover counts (± s.e.) across learning blocks (1-3). Block 4 shows the results of removing or changing emphasized objects.

Figure 7: Hover count results after block 0 normalization

NAM-12 (9964ms) and NAM-9 (11905ms), both of which were located at or near the corners. We also found the hardest targets were those located in the centre, such as NAM-18 (19295ms) and NAM-2 (18345ms). Although previous research suggests that targets such as NAM-18 and NAM-2 should have been much more difficult in the *No-Landmarks* condition (where there were no visual features to help users remember these locations) [58], search times actually favoured the *No-Landmarks* condition. Even with target NAM-63 – which was highlighted in the landmark conditions – we found that participants in the *No-Landmarks* condition found the target faster (9919ms) than those in the landmark conditions (15103ms).

## 4.2 S1 Effects of Change/Removal of Landmarks: Search time, Hovers, and Errors Block 3 to 4

### 4.2.1 S1 Change/Removal – Search Time:

To look for effects of removing/changing the accidental landmarks, we carried out an analysis using only block 3 (the block before the removal/change) and block 4 (the block after the removal/change). The 3x2 RM-ANOVA (*Condition* x *Block*) found an interaction between the two factors ($F_{2,178} = 3.43, p = 0.03, \eta^2 = 0.02$) in terms of search time.

Search times increased in *Landmarks-Removed* from 13472ms in Block 3 to 15034ms in block 4, and in *Landmarks-Changing* from 11830ms to 13774ms. By contrast (and as expected from previous literature on learning [10]), search performance continued to improve in the *No-Landmarks* condition: from 10128ms in Block 3 to 7768ms in Block 4. To check whether each condition changed significantly between block 3 and 4, we carried out additional follow-up t-tests. However, no specific differences were found for this per-condition analysis (all $p > 0.05$).

Following the analysis on the learning blocks presented above, and the significant interaction between *Condition* x *Block* for Blocks 3 and 4, we carried out an analysis of the final block using a similar 3x7 RM-ANOVA (*Condition* x *Block*). We found a main effect of *Condition* ($F_{2,178} = 12.61, p < 0.001, \eta^2 = 0.04$) and *Target* ($F_{6,522} = 7.84, p < 0.001, \eta^2 = 0.04$) on search times, and an interaction between *Condition* x *Target* ($F_{12,522} = 1.84, p < 0.001, \eta^2 = 0.02$). Post-hoc pairwise t-tests again showed significant differ-

ences between *No-Landmarks* and both landmarks conditions (both $p < 0.05$).

### 4.2.2 S1 Change/Removal – Hovers:

To look for effects of removing/changing the landmarks on hover count, we carried out a similar analysis using hover data from block 3 and block 4. The 3x2 RM-ANOVA (*Condition* x *Block*) found an interaction between *Condition* x *Block* ($F_{2,178} = 3.49, p = 0.03, \eta^2 = 0.01$). Hovers increased by in *Landmarks-Removed* from 7.75 to 8.97 and in *Landmarks-Changing* from 6.64 to 7.9. As with search time, performance continued to improve in the *No Landmarks* condition: from 7.1 hovers in Block 3 to 5.7 in Block 4. We carried out additional follow-up t-tests to check whether each condition changed significantly between block 3 and 4, however, no specific differences were found for this per-condition analysis (all $p > 0.05$).

Following the significant interaction between *Condition* x *Block*, we carried out an analysis of the final block using a similar 3x7 RM-ANOVA (*Condition* x *Block*). We found main effects of *Condition* ($F_{2,178} = 3.28, p = 0.04, \eta^2 = 0.04$) and *Target* ($F_{6,522} = 7.22, p < 0.001, \eta^2 = 0.05$) on hovers, and an interaction between *Condition* x *Target* ($F_{12,522} = 2.04, p < 0.001, \eta^2 = 0.02$). Post-hoc pairwise t-tests showed significant differences between *No-Landmarks* and both landmark conditions (both $p < 0.05$).

### 4.2.3 S1 Target-by-Target Analysis:

As the ANOVA results for the final block showed a main effect of *Target* on search time and hover counts, and also showed interactions between *Condition* and *Target*, we again looked into the results of each specific target.

The *Condition* x *Target* interaction indicates that the effect of *Condition* on search time and hover count varied by target. Inspecting the target-by-target charts shows that there were no targets for which the landmarks were particularly helpful during training, and that the majority of targets were affected by the removal of the landmark. To explore this further, we repeated an ANOVA for each target in Block 4, to see which targets were affected by *Condition*. For Search times, the following targets showed significant effects of Condition: NAM9 (p=0.04), NAM18 (p=0.03), NAM32 (p=0.01), and NAM63 (p=0.01). For hover count, only NAM9 showed a significant effect of condition (p=0.02). In all these cases, search times and hover counts were significantly better in the *No-Landmarks* condition.

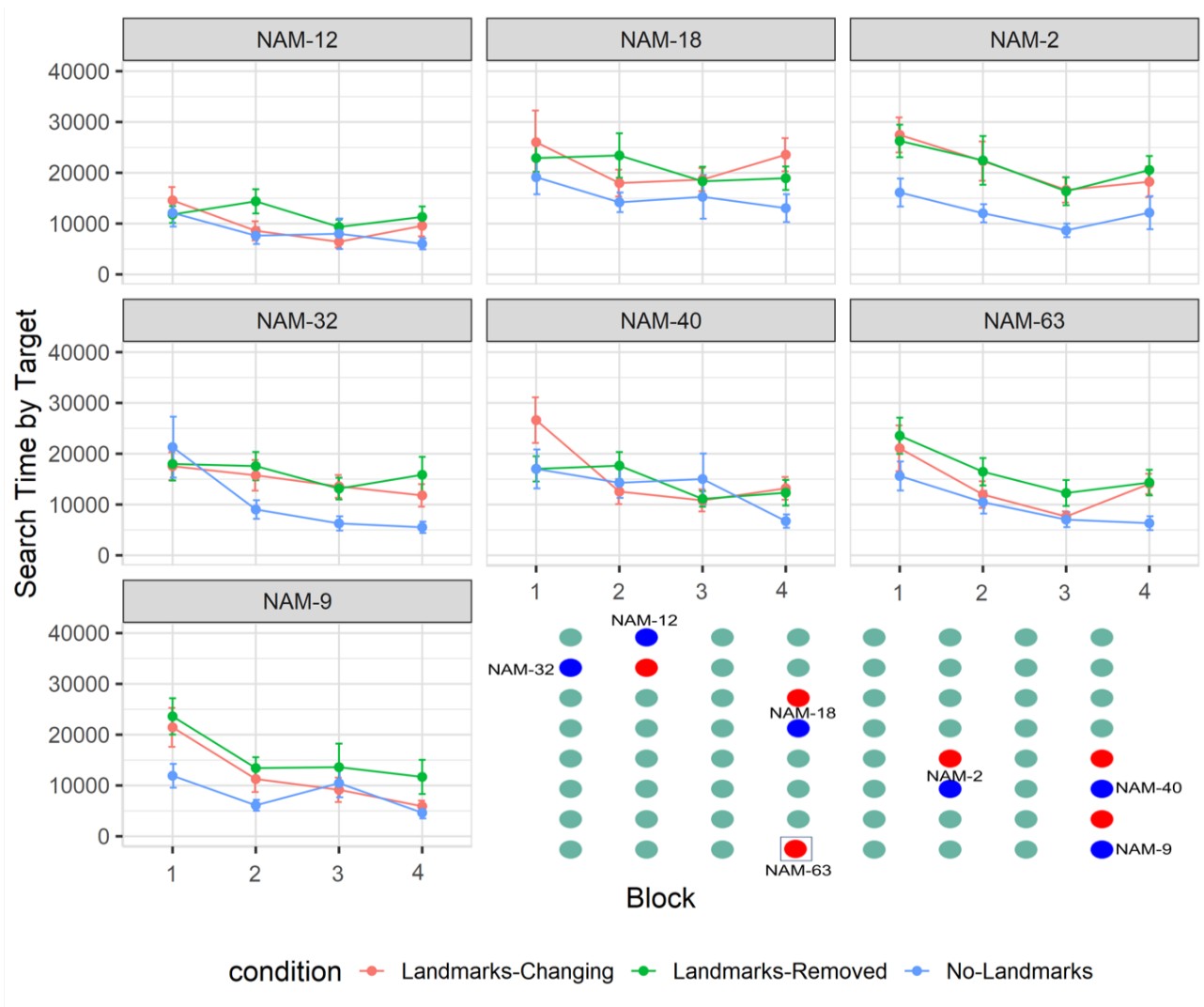

Figure 8: Mean search time ($\pm$ s.e.) by Target and Block

Similar to the learning blocks, we found that targets located near the corners required the fewest hover actions and were found fastest: NAM-12 (8926ms) and NAM-9 (7361ms). The most difficult targets were those located in the centre: NAM-18 (18456ms) and NAM-2 (16945ms). For NAM-63, which was located directly on a landmark during training, we found that participants in the *No-Landmarks* condition on average found the target twice as fast (6328ms) as those in the landmark conditions (14197ms).

### 4.3 S1 Subjective Measures

#### 4.3.1 S1 Perceived Change in Difficulty

For each specific target, we also asked participants in the landmarks conditions to rate on a 1-7 scale how much more difficult it was to find the target in block 4 compared to block 3. (We did not ask this question to participants in the No-Landmarks condition, but we can assume that they would not have seen any major difference in difficulty between block 3 and 4). Mean results are shown in Fig 10. Overall, the target that participants felt was least affected by changing or removing the landmarks was NAM-9, located in the bottom right corner of the grid.

Using an Aligned Rank Transform on the difficulty ratings [62], one-way ANOVAs were performed for each of the targets using Condition as the factor. The ANOVA found a significant effect

of Condition for the NAM-40 target (middle of the last column in the grid): for this target, participants in the *Landmarks-Removed* condition rated the target as more difficult (4.65) than participants in the *Landmarks-Changing* condition (3.8).

#### 4.3.2 S1 Perceived Effort

Participants' perceived effort was recorded using the NASA-TLX questionnaire [25]. For the *Landmarks-Changing* condition and the *Landmarks-Removed* condition we specifically asked the effort questions in relation to their perceived effort after the landmarks were changed or removed. We used an Aligned Rank Transform on the aggregated responses to perform a one-way ANOVA on each of the TLX questions using Condition as a factor. The mean responses to the TLX questions are shown in Fig 11. Significant effects were found in the responses for *perceived success* and *frustration*, both ($p < 0.05$). Holm-corrected post-hoc pairwise t-tests were performed on the questions that had significant effects. For perceived success, the pair-wise comparison found a significant difference between *Landmarks-Removed* and *No-Landmarks*, with participants having a greater perceived success with no landmarks rather than when landmarks are initially present and then removed.

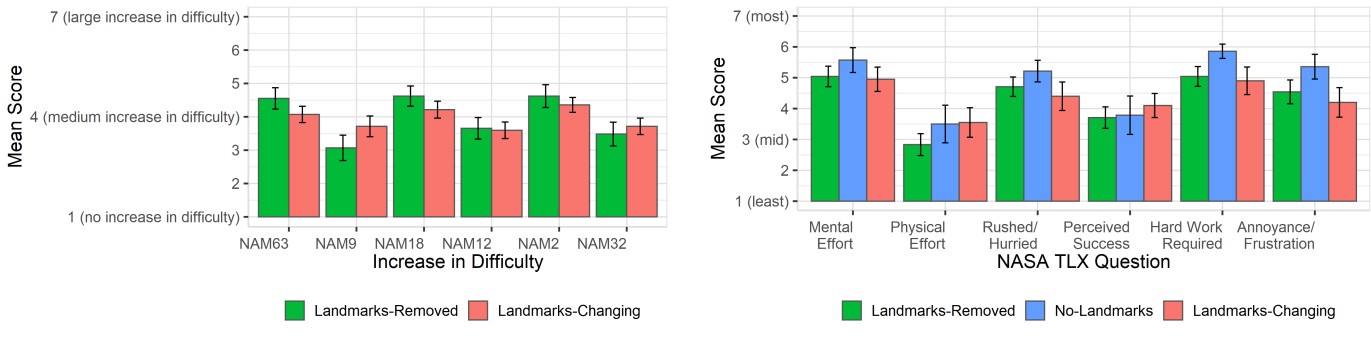

Figure 9: Mean hover count ($\pm$ s.e.) by Target and Block (hovers counted after 300ms)

Figure 10: S1 Perceived Change in Difficulty in Block 4

Figure 11: S1 Mean NASA Task Load Index scores, by condition

### 4.3.3 S1 Participant Comments

At the end of the study, we asked participants to explain their general process of finding targets and whether some were easier or harder than others. Their responses generally echoed several of the findings from the previous sections. While search times and hover counts did not show a clear improvement in conditions with landmarks,

some of the participant's remarks do state the benefit of having landmarks. For example, P1 stated "When they [targets] were close to a different colored circle, in the immediate vicinity, it made it easier." Similarly, P3 stated "For some of the targets, I was able to find them easily because they were near a red colored circle." A few other participants also remarked how specific targets were easier,

such as P8 emphasized, "NAM-9 in the corner, [and] it was [easier] around the colored ones."

For participants in the *No-Landmarks* condition, comments suggested that people had to resort to other techniques: for example, P10 stated "I used Cartesian indication (from high school; 2 axes: x and y)." P12 said, "32 was in the first column second row which was easy to find, 9 was at the end and 63 was at the third column last raw" and P13 stated "corner, first and last row."

# 5 STUDY 2: EFFECTS OF ACCIDENTAL LANDMARKS IN MORE-COMPLEX SCATTERPLOTS

Our second study considers visualizations that are more complex than the grid used in the first study. Visualizations often have an irregular layout (e.g., scatterplots where designers have no control over how the data is organized) that may contain more structural and layout-based visual features that can act as landmarks – such as clusters of points and white space, in addition to edges and corners. We need to understand how users perceive landmarks in more complex visualizations, so to increase the complexity of the task (compared to Study 1) we designed our study task to show a scatterplot visualization based on a real-world dataset from the Gapminder site [43]. Utilizing a real-world dataset ensured we would have little control over where and how the potential relationship between the targets and the accidental landmarks (and any other visual features that can act as landmarks) would be formed.

## 5.1 S2 Study System and Targets

A web-based system was developed using JavaScript and D3 [7] that showed a scatterplot based on Gapminder data [43]. The X axis showed the per-capita income of a country, and the Y axis showed the life expectancy in that country for a single selected year (similar to other online recreations of the dataset). The interface presented the names of target countries, and the user had to click on a target item to confirm a selection.

For the study we used seven targets, out of 142 total items on the screen; eight items were initially highlighted as accidental landmarks. The seven targets were chosen to have different inherent difficulties based on their proximity to landmarks and other potential spatial cues in the visualization (e.g., edges, clusters, and white space). Targets and locations are shown in Figure 12.

## 5.2 S2 Procedure

We followed a similar procedure to Study 1, with seven phases: (1) informed consent, (2) demographics questionnaire, (3) vision test, (4) guided tour, (5) study tasks, (6) post-study questionnaires, and (7) debriefing. As with Study 1, participants first completed a guided tour through all the targets after which the study proceeded to the test phase.

## 5.3 S2 Participant Recruitment

We initially recruited 90 participants across the three conditions (30 per condition) using Amazon's Mechanical Turk (MTurk), and gathered data through a custom browser-based experiment tool [31]. Three participants were removed for having an overall completion time for the study over 3 SD from the mean, and an additional participant was removed due to completing experimental tasks more than once (refreshing the browser causes the tasks to restart). The remaining participants were distributed as follows: 30 in *Landmarks-Removed*, 28 in *Landmarks-Changing* and 28 in *No-Landmarks* ($\mu_{age}$ = 35.47, $\sigma_{age}$ = 12.11, 55 men, 30 women, 1 preferred not to answer). Our study required workers to have over 90% HIT acceptance rate, and we also checked the questionnaire responses to ensure that the same answer was not used for all of the questions, as well as whether the study was completed too quickly or too slowly (which could represent participants simply clicked through the study, or were focused on additional tasks).

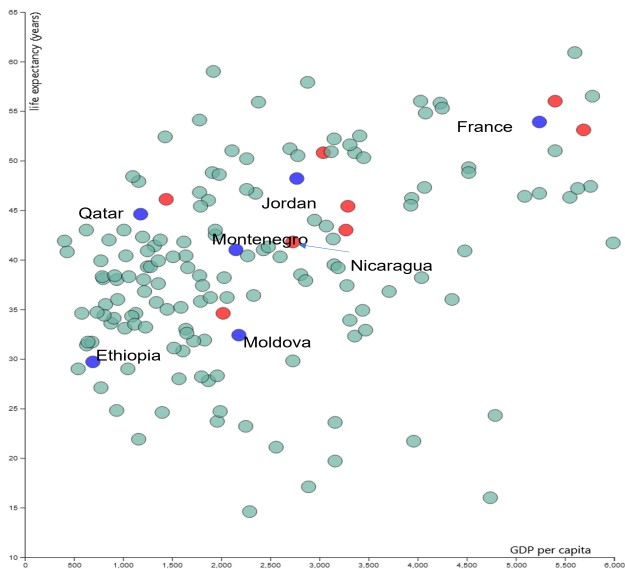

Figure 12: Locations of the targets (shown here in blue, not shown in the study) in relation to the landmarks. Target Nicaragua (shown with a red dot) was both a target and a landmark. In the *No-Landmarks* condition, there were no red highlights.

All participants were paid $3USD for completing the study, which took approximately 15 minutes.

## 5.4 S2 Study Design

The goal for this study was to understand the effects of landmarks on spatial learning in more-complex visualizations. Our main research questions (RQ) for this study were:

- RQ-1: Do landmarks improve finding and re-finding targets in scatterplots when they are present (i.e., decreased search time, number of hovers, and error rate)?

- RQ-2: Does removing or changing landmarks after a learning period affect re-finding (i.e., increased search time, hover counts, and error rate)?

To investigate these questions, the study used a mixed factorial design with three factors:

- Condition (between-subjects): *No-Landmarks*, *Landmarks-Removed*, *Landmarks-Changing*

- Target Locations (within subjects): seven target locations (see Figure 12)

- Blocks (within subjects): 1-4 (blocks 1-3 are training; block 4 removes/changes any landmarks).

Similar to Study 1, our dependent variables were search time, hover counts (only included if longer than 300ms), and error counts (i.e., incorrect clicks), and subjective ratings of effort from post-session questionnaires. Targets were the same for all participants.

# 6 S2 STUDY RESULTS

We again report effect sizes for significant ANOVA results as generalized eta-squared $\eta^2$ (considering .01 small, .06 medium, and >.14 large [36]). Outliers were determined as any trial with a search time greater than 3 SDs above the block's mean. 85 of the 2408 trials were removed from the analysis. All pairwise t-tests were corrected using the Holm-Bonferroni method.

## 6.1 S2 Effects of Landmarks on Learning: Search Time, Hovers, and Errors in Learning Blocks

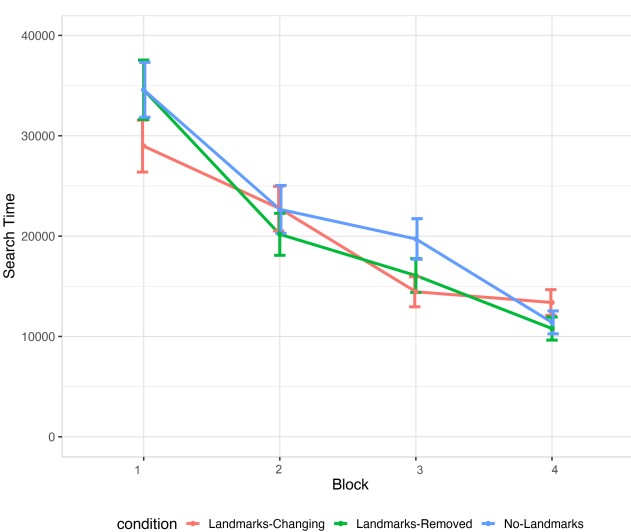

Figure 13: ScatterPlot Mean trial search times (± s.e.) across learning blocks (1-3). Block 4 shows the results of removing or changing emphasized objects.

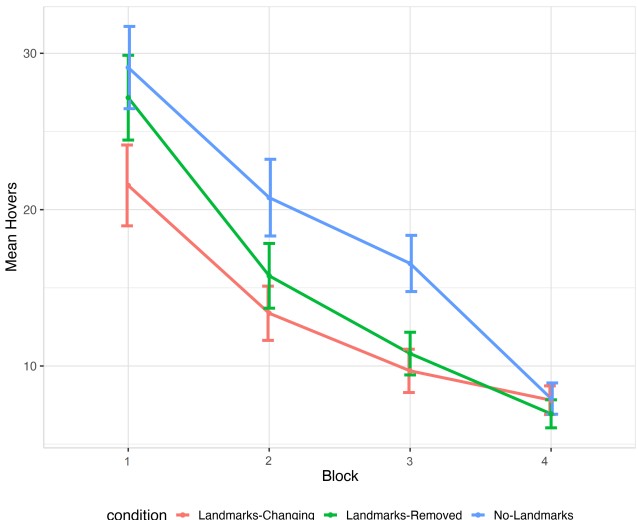

Figure 14: ScatterPlot Mean Hovers(± s.e.) across learning blocks (1-3). Block 4 shows the results of removing or changing emphasized objects.

### 6.1.1 S2 Learning Blocks – Search Time

Search time in the test trials was measured from the time a target name appeared on the screen to the time a user correctly found a target. Search times across all blocks for the three conditions are shown in Fig 13.

For the learning blocks, a 3x3x7 RM-ANOVA (*Condition* x *Block* x *Target*) showed a main effect of *Condition* ($F_{2,172} = 4.99, p = 0.007, \eta^2 = 0.005$) *Block* ($F_{2,172} = 70.06, p < 0.001, \eta^2 = 0.07$) and *Target* ($F_{6,510} = 7.72, p < 0.001, \eta^2 = 0.02$). Aggregated across all training blocks and targets, participants in the *Landmarks-removed* took 23367ms to find a target, 21907ms in the *Landmarks-Changing* compared to an average of 25339ms in the *No-Landmarks* condition. Post-hoc pairwise t-tests showed a significant difference on search times between *Landmarks-Changing* and *No-Landmarks* (p=0.008) for the learning blocks.

### 6.1.2 S2 Learning Blocks – Hovers

We again measured the number of hovers as the number of times the participant held the cursor over an element for 300ms or more to show the name. Mean hovers per trial are shown in Figure 14. For the learning blocks, a 3x3x7 RM-ANOVA (*Condition* x *Block* x *Target*) showed a main effect of *Condition* ($F_{2,172} = 8.92, p < 0.001, \eta^2 = 0.006$), *Block* ($F_{3,255} = 52.57, p < 0.01, \eta^2 = 0.06$) and *Target* ($F_{6,510} = 12.26, p < 0.001, \eta^2 = 0.02$). On average (across all learning blocks), it took a participants 18.25 hovers to find a target in the *Landmarks-Removed*, 22.24 hovers in the *Landmarks-Changing*, while the *No-Landmarks* required 28.48 hovers for a correct selection. A Post-hoc pairwise t-test showed a significant difference on Hovers between both *Landmarks* conditions and *No-Landmarks* (both $p < 0.01$) for the learning blocks.

### 6.1.3 S2 Learning Blocks – Errors

Errors were measured as the number of incorrect clicks before choosing a correct target. Overall, there was an average of 0.69 errors per trial across all conditions and blocks. For the learning blocks,

a 3x3x7 RM-ANOVA (*Condition* x *Block* x *Target*) showed a main effect of *Condition* ($F_{2,170} = 4.13, p = 0.01, \eta^2 = 0.003$) and *Block* ($F_{3,255} = 4.51, p = 0.003, \eta^2 = 0.006$) errors, but no interactions between the factors. A Post-hoc pairwise t-test showed significant differences between *Landmarks-Changing* and *No-Landmarks* (p=0.01), with participants making fewer errors with the *Landmarks-Changing* conditions overall (0.50 errors per trial) compared to 2.37 errors per trial for the *No-Landmarks* condition.

### 6.1.4 S2 Learning Blocks – Target-by-Target Analysis

As the ANOVA results showed a main effect of *Target* on search time and hover counts, we looked into the results of each specific target. Although there is little difference among most targets, targets near the centre (such as Jordan and Montenegro) were the hardest to find (see Figures 15 and 16). Nicaragua (which was both a target and highlighted in the landmark conditions) was substantially harder to find in the *No-Landmarks* condition than the rest of the targets (required 42.5 hovers and 60900ms), but was only of average difficulty in the landmarks conditions (19.85 hovers and 31900ms). In all conditions, France, located near the top right corner (between landmarks in the landmark conditions) was the easiest to find (13364ms and 6.9 hovers).

## 6.2 S2 Effects of Change/Removal of Landmarks: Search time, Hovers, and Errors Block 3 to 4

### 6.2.1 S2 Change/Removal – Search Time

To investigate the effects of changing or removing landmarks in the scatterplot, we carried out an analysis using only block 3 (the block before the removal/change) and block 4 (the block after the removal/change). The 3x2 RM-ANOVA (*Condition* x *Block*) did not find an interaction between the two factors (p = 0.054) for search time. There was also no interaction between (*Condition* x *Target*) (p=0.54).

Search times continued to decrease in *Landmarks-Removed* from 16072ms in Block 3 to 10788ms in block 4, and in *Landmarks-Changing* from 14454ms to 13385ms. These results were similar to the *No-Landmarks* condition: from 19716ms in Block 3 to 11401ms in Block 4. However, this improvement varied by target, and performance got worse for some items: for example, Nicaragua was both

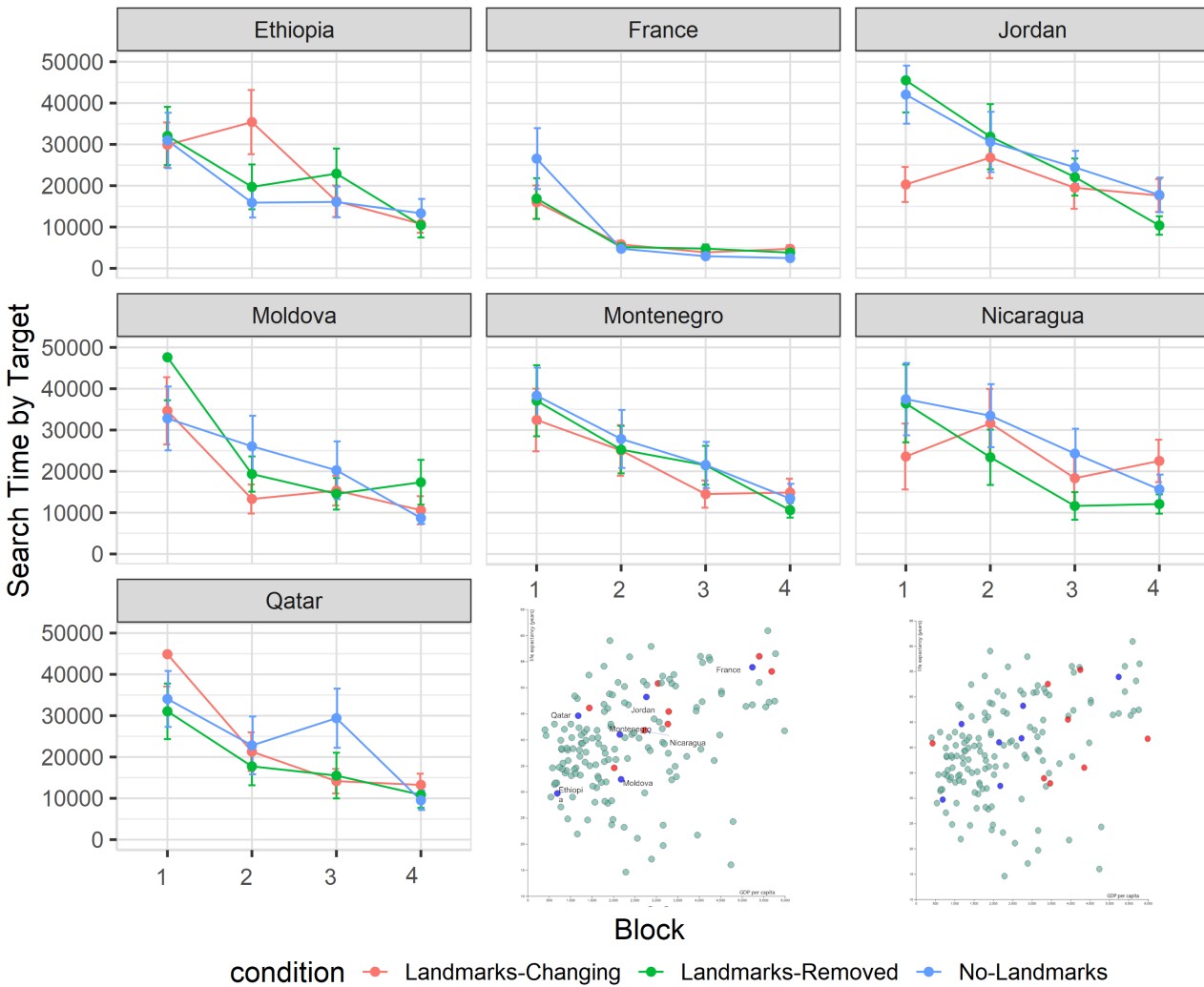

Figure 15: Mean search time (± s.e.) by Target and Block in Scatterplot Study. Target/landmark locations (and the change in Landmarks) are included in the bottom right corner.

a landmark and a target, and for this target search times went from 18384ms to 22522ms in the *Landmarks-Changing* condition. We saw a similar (although smaller) effect with France and Montenegro.

### 6.2.2  S2 Change/Removal – Hovers

We did a similar analysis investigating the change or removal of landmarks for hovers. Similar to search times, we saw an improvement in hovers required to find a target for all conditions, going from 9.6 hovers to 7.8 for *Landmarks-Changing*; 10.79 to 6.93 in *Landmarks-Removed*, and 16.5 to 7.97 for *No-Landmarks*. The 3x2 RM-ANOVA (*Condition* x *Block*) found an interaction between *Condition* x *Block* ($F_{2,172} = 3.73, p = 0.002, \eta^2 = 0.006$) for Hovers.

Similar to search times, hover counts continued to decrease from the 3rd to 4th block, but certain targets were affected negatively. We saw the same effect for Nicaragua (going from 9 hovers in Block 3 to 13 when landmarks were changed— but this effect did not happen when the landmark was removed). Conversely, Moldova was negatively affected by the removal of landmarks (from 8 hovers in Block 3, to 10 in Block 4), but not by changing the landmarks (continued to improve to just 3.14 hovers in the final Block).

### 6.3  S2 Subjective Measures

#### 6.3.1  S2 Perceived Change in Difficulty

For each specific target, we also asked participants to rate on a 1-7 scale how much more difficult it was to find the target in block 4 (for the landmark conditions). As shown in Fig 17, while participants did report that the change/removal of landmarks made the task more difficult, the change affected most targets equally. Overall, France (located top right corner between two red dots) was the least affected by the change. A one-way ANOVA on each of the targets ratings using Aligned Rank Transform [62] with Condition as the factor found no differences between the conditions.

#### 6.3.2  S2 Perceived Effort

We again measured participants' perceived effort in relation to changing or removing the landmarks was recorded using the NASA-TLX questionnaire [25]. For the *No-Landmarks* condition, perceived effort relates to finding the target in the final block. Results are summarized in Fig 18. We used an Aligned Rank Transform on the aggregated responses to perform a one-way ANOVA on each of the TLX questions using Condition as a factor. The ANOVA found no significant differences between the conditions on any of the TLX measures (all $p > 0.05$).

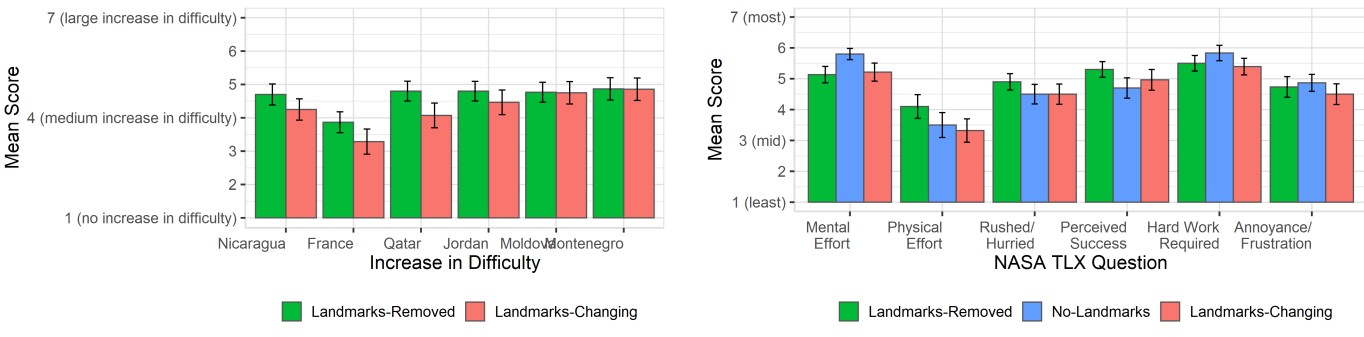

Figure 16: Hovers (± s.e.) by Target and Block in Scatterplot Study. Target/landmark locations (and the change in Landmarks) are included in the bottom right corner.

Figure 17: S2 Perceived Change in Difficulty in Block 4

Figure 18: S2 Mean NASA Task Load Index scores, by condition

### 6.3.3 S2 Participant Comments

At the end of this study, we also asked participants to explain their overall process of finding targets and whether they employed any specific strategy throughout the task. Regarding the scatterplot configuration and using a real dataset, P8 stated "Yes, some targets

[were easier as they] were located near a corner, or distinct cluster" while P12 commented, "some [targets] were located next to same continent countries or countries near by" while P44 mentioned, "I tried to remember some of the countries in a certain area of dots whose names I am familiar with." Other participants mentioned the

use of landmarks, such as P32 stated, "[targets were easier] only when they were within the red circles" and P90 mentioned, "Some [targets] were close to the edges and red circles." Participants in the *No-Landmarks* condition more commonly stating using their personal experience to help with the task, such as P2 mentioning "I noticed countries that are close together on a map were relatively close together on this chart", and P79 "I found middle east countries, mostly all together, and European and Asian countries were similarly grouped, and from there I just had to try to build a memory."

## 7 DISCUSSION

Our studies investigated whether accidental landmarks could help users find and re-find items in a visualization, and whether they impaired performance when taken away or changed. The studies provide several main findings:

- In Study 1, performance in the learning blocks (in terms of search time and hovers) was no better for the accidental-landmark conditions (in fact, the *No-Landmarks* condition was best), but in the final block, performance was impaired when landmarks were removed or changed;

- In Study 2, search time and hovers in the learning blocks were lower for the accidental-landmark conditions, but there was no significant detriment in the final block when the landmarks were removed or changed;

- In both studies, participants in the landmark conditions reported that finding targets in the final block was more difficult than in the previous blocks;

- In both studies, participant comments suggested that people were using the highlight colours to assist them in finding the targets, as well as structural landmarks such as corners, edges, clusters, and white space;

- In both studies, participants reported no major differences between the three conditions in terms of overall effort.

In the following sections, we provide explanations for these results, discuss how our findings can generalize to real-world visualizations, and outline limitations of the study and opportunities to extend the research.

### 7.1 Explanation of Results

#### 7.1.1 The Effects of Landmarks in Learning Blocks

Our two studies showed contrasting results about the usefulness of accidental landmarks in helping participants learn item locations during the learning blocks. The only change between the studies was in the type of visualization used, and the differences between the grid and scatterplot can help to explain the contrasting study results. First, in the simple grid used with Study 1, the visual search task was easier than with the more complex scatterplot of Study 2. Study 1 participants could carry out a row-by-row or column-by-column search pattern to look for the target, which may have made the coloured highlights less valuable. In contrast, the irregular and more complex organization of items in the Study 2 scatterplot did not allow users to carry out a methodical search strategy, and when users are carrying out a less-organized search, the anchors provided by the highlighted landmarks may have been more valuable. For example, a general problem in searching a complex dataset is that users repeat some areas and miss others; the reference frame provided by the highlights may have assisted users in organizing their search and reducing repetition.

Second, the attentional draw of the emphasized points may have affected the two visualizations differently. It is known that bottom-up attention will be guided to areas of visual emphasis (e.g., our studies showed the highlights as red circles among white circles) [16, 64]. Visual attention will be in part guided to objects that differ from others as a first step in the multi-step process of attention, which is then guided by the task and previous selections. In the simpler grid visualization, the attentional draw of the emphasized points may have distracted participants from a regularized search strategy, reducing the efficiency of their visual search. Although this could also have occurred in the more-complex scatterplot of Study 2, any negative effects may have been outweighed by the organizational benefit provided by the reference frame of the coloured landmarks.

These possibilities should be explored further in additional studies. In addition, we also note that our between-participants design leads to the potential for inherent group differences that may account for some of the overall difference between conditions during training. It was not possible to completely remove these group differences (e.g., we could not use performance on the first block as a covariate, because the experience of visual search was substantially different for the landmarks and no-landmarks conditions); further studies can help to further investigate the initial differences between the conditions.

#### 7.1.2 The Effects of Removing / Changing Landmarks

Our studies also showed contrasting results in terms of whether changing or removing landmarks impaired performance: Study 1 saw a significant reduction in performance when landmarks were taken away or changed, whereas Study 2 did not (there were indications of a performance reduction for some targets, but not overall).

Again, differences between the grid and scatterplot visualizations can help to explain these contrasting results. In Study 1, the relative lack of structural or layout-based landmarks in the grid means that the coloured landmarks were more likely to be seen as a primary reference frame for participants in the landmarks conditions (particularly because people were not forewarned that the highlights would be changed / removed). For example, Study 1 saw strong performance impairments for both targets that were in the interior of the grid (near to a coloured highlight but not near to a corner or an edge).

The scatterplot used in Study 2 had many more structural and layout-based landmarks in addition to the coloured highlights (e.g., clusters of points and areas of white space in addition to the edges and corners of the datapoints). This means that participants in Study 2 had multiple frames of reference available to them, and they likely made use of both structural and colour-based landmarks when learning item locations. Previous research suggests that people will use whatever reference frame makes their task easiest, but in Study 2, neither reference frame was dominant. There were eight highlighted items in the Study 2 scatterplot, meaning that the coloured items did not simplify the task so much that it was trivially easy (e.g., the task was much more difficult than if there had been two targets that were beside two coloured landmarks). The overall difficulty means that participants were likely to make use of the structural landmarks in addition to the highlighting – and since structural landmarks were unchanged in the final block, people may have been able to rely on this other reference frame to maintain their performance. Limited evidence for this hypothesis can be seen in the performance of the Nicaragua target – because this target was also highlighted in the training blocks, it was easy to find using only colour, which may have led participants to rely more on colour rather than structural landmarks such as nearby clusters.

Overall, our results align with the guidance and effort hypotheses (i.e., that providing guidance and reducing effort in training will lead to over-reliance on the guide). When colour highlighting was the only reference frame available, or when it made the retrieval task easier, participants relied on it more and had larger reductions in performance when the highlighting was removed or changed. The presence of other reference frames (e.g., structural and layout-based

landmarks) appeared to mitigate the problems caused by removing the colour highlights – but it is worth noting that in Study 2, participants in the landmarks conditions subjectively rated the task as substantially more difficult when the landmarks were removed or changed, even though they were able to make use of other knowledge to preserve performance.

## 7.2 Generalizing the Findings to other Contexts

Our study examined the effects of accidental landmarks in two visualization settings, a simple grid and a more-complex scatterplot, and there are several underlying commonalities between our experiments and real-world scenarios that argue for the generalizability of our findings.

First, our learning task – repeatedly visiting target locations – is common in many real-world visualization tasks. A typical exploration of a dataset involves investigating interesting data points or patterns to identify relationships between them. Additionally, it is common for visualization designers to use emphasis to encourage exploration (e.g., by highlighting regions of interest to signify importance or to alert viewers to missing links). Similarly, in narrative visualization, when known aspects of a data set are presented to the viewers [8, 49], different data points are explained and presented to viewers, and designers may alter an element's size or colour to improve its legibility relative to other areas of a visualization, potentially making it more memorable.

Second, our manipulation of the landmarks – changing the emphasized set or removing emphasis altogether – is also something that is likely to occur in many real-world visualizations. Emphasizing or highlighting one particular subset of the displayed data is a common action as viewers explore different aspects of a visualizations or review different findings. As the exploration or story-telling process continues, it is common for users to focus on a different subset of the data. For example, in Study 2, a normal exploration process could involve highlighting countries in different continents. Once an analysis or exploration session is finished, unless the visualization system has a history mechanism built in, there will be no emphasized points upon returning to a visualization (similar to our *Landmarks-Removed* condition).

Third, our participants were MTurk workers rather than users who have naturally arrived at a visualization task, and although there are likely to be differences between these populations in terms of intrinsic motivation and interest in the dataset, there are also many similarities. In particular, there is a wide range of visualization users who could be affected by accidental landmarks, and the demographics of our MTurk sample covered a variety of prior experience with visualizations. The characteristics of an MTurk study help increase ecological validity compared to more typical lab studies: we had a larger sample than typical laboratory studies (180 total participants) who had a much more diverse background than what is generally seen in HCI experiments, as such our findings can be more representative of a generalized user base.

Fourth, our use of emphasis in the studies reasonably represents the type of accidental landmarks that may be available in a visualization system – e.g., highlight-based filtering and brushing capabilities are now common in many tools, such as Tableau as shown in Figure 1 – and many users will take advantage of these capabilities.

## 7.3 Limitations, Extensions, and Future Work

There are limitations to our evaluation – many of which were necessary to test the use of emphasis as landmarks in controlled environments – and these limitations provide opportunities to expand our work in future studies.

The grid-style visualization, the underlying dataset (target/distractor names), and the target/landmark locations were chosen for the study in order to control potential external factors such as

cluster-based layout cues that provide visual indications about location. As our grid with circles most resembles a scatterplot, we then extended our initial results to evaluate the effects of emphasis on spatial memory using a scatterplot. However, we used a single dataset behind the scatterplot, and we note that participants may have formed relationships within the scatterplot and dataset (familiar names or clusters of data). This can be counteracted by evaluating multiple distinct datasets in followup work.

Second, our future work involves evaluating the use of landmarks in a greater variety of chart types including bar charts or more complex, interactive visualizations (e.g., basic charts in a small-multiples configuration). Involving multiple charts may result in the benefits or drawbacks of landmarks being amplified as there may be more structural landmarks occurring on the outlines of multiple charts, but it may be harder to find items within each chart.

Third, we explored the effects of accidental landmarks with only one visual variable (colour), but there are many other emphasis effects that could be tested, including size, outline, transparency, texture, or shape. Previous research has shown that different visual variables attract attention and affect learning at different levels [9, 22, 38], and designers must decide on a trade-off between noticeable highlights and the potential unintended distraction in learning.

Fourth, our study focused on immediate learning performance and short-term memory and spatial awareness through our revisitation task; we did not test longer-term retention after hours or days (which would be common in visualization as analysts can work with datasets over extended periods of times of weeks and months). Our approach was necessary to establish an initial baseline understanding of how emphasis affects the initial spatial learning process, but in future studies we will extend the work to look at longer retention. Furthermore, development of real expertise with a visualization system often requires a much longer training period than what was provided by our studies. In future work, longer-term studies will allow us to examine how extended training periods and varying gaps of hours or days can lead to better spatial development and retention.

In addition, there are several research directions that could explore ways of better supporting users even when accidental landmarks change or disappear. Our results and participant comments show that users do use and rely on highlights to revisit previous targets, particularly when the landmarks make the retrieval task easier. Even though designers cannot control the application of filters and highlights when users explore a visualization, there may be ways of avoiding the problems that can arise from changes to emphasis and highlighting. One possibility is to show traces of previous highlights (i.e., "ghost echos" or "phosphor effects"); these marks would provide assistance to users who are relying on accidental landmarks, by providing at least a trace of the landmarks' previous locations. These traces could slowly fade away after a period of time, which could also encourage users to find other strategies for remembering the items.

Further study is also needed on the general problem of supporting revisitation, and whether other mechanisms that could be used to improve re-finding can also act as accidental landmarks. For example, "visit wear" techniques can visually mark the items that people visit in a visualization, making revisitation much easier. An example of this technique is the Footprints scrollbar, which records user locations with marks in a scrollbar if the user pauses for more than one second [1]. This system also analysed usage data to improve and automate the state saving algorithm such that the most relevant locations would be saved without cluttering the scrollbar. While visualizations can range from very simple representations to very complex multi-dimensional parameter spaces, a combination of methods such as visit wear and state saving mechanisms can ease revisiting objects while exploring visualizations. Further work is needed to understand whether and how annotations such as visit-wear marks function as landmarks, and whether their obvi-

ous value in supporting revisitation can lead to larger problems of over-reliance.

## 8 CONCLUSION

Many visualizations display large datasets in which it can be difficult for users to find (and re-find) specific items. Interactive systems that provide highlighting tools such as filtering or brushing emphasize certain data points – and these can become "accidental landmarks," visual anchors that help users remember locations that are near the emphasized points. Landmarks are known to be useful (by aiding revisitation), but previous research on the guidance hypothesis suggests that if users become dependent on them, removing or changing the highlighting could cause problems. We provide designers with new information about these issues: we carried out two crowd-sourced studies, first in a basic grid configuration and then in a traditional scatterplot, in which people were asked to learn a set of item locations with or without emphasized points. We then removed or changed the highlighting to see if performance suffered. Results show that accidental landmarks did not improve performance during training in a basic grid, but did so for a scatterplot, and changing or removing emphasized data points affected users' ability to re-find targets – particularly those that were not near structural landmarks such as the corners of the visualization. Our work provides new knowledge about how visual features, emphasis, and landmarks in visualizations can affect revisitation, and new understanding for designers who want to support spatial awareness and learning in visualizations.

### ACKNOWLEDGMENTS

We thank the anonymous reviewers for their feedback, the members of the Interaction Lab at the University of Saskatchewan for their continued support and our participants.

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
