# OpenReview forum: "Accidental Landmarks: How Showing (and Removing) Emphasis in a 2D Visualization Affected Retrieval and Revisitation"
_graphicsinterface.org/Graphics_Interface/2022/Conference — GI 2022_

### Official Review · Reviewer_sq42 · 2022-04-13
**A thorough and interesting investigate into the core HCI topic of landmarks in visualization.**

**Rating:** 9
**Confidence:** 3

**Review:**

The authors explore the effects of accidental landmarks in visualizations that have more or less inherent structure.
They find that, if there is existing strong structure (e.g., a grid layout) to the data, accidental landmarks do not help and may even hinder search.
For complex, less structured data (in the paper, a scatterplot), accidental landmarks are useful, but are less useful if they change (e.g., filtering systems).

Pros:
- Thorough exploration of a topic
- pros and cons with mixed methods providing quantitative and qualitative results for future designers
- good methodology.

Cons:
- qualitative data was weak with poor method reporting.

Overall this was a refreshing return to core HCI for me. It was a strongly written and informative paper, with a good methodology and clear and interesting results.
In particular, I enjoyed all the figures that presented the data in interesting ways. I especially found the breakdown per target to be interesting and an insightful analysis decision, essentially understanding that the visualization itself can provide structure in which to scaffold learning alongside the landmarks.

In addition, I found the discussion worthwhile, and led me to have many questions of my own for future research.

My only real criticism is not very strong, and it's that I found the qualitative reporting of participant comments to be weak, and perhaps I missed it, but I was unclear how these were analyzed. As these were more used for adding texture to the results and do not inform major recommendations or results, I think it is fine to leave as a point of reference for future research questions.

Unfortunately, weak effects, but clear results otherwise with some interesting lessons. I think this paper should be accepted. Well done!

---

### Official Review · Reviewer_uMER · 2022-04-14
**The paper introduces an interesting visualization concept.**

**Rating:** 6
**Confidence:** 4

**Review:**

The paper introduces the concept of “accidental landmark” to support revisitation in a visualization representation.
The idea behind this “accidental landmark” is interesting and has a lot of potentials. Two crowdsourced studies were conducted to evaluate the effect of this phenomenon; however, no significant impact was found on search time or error rate. This could be due to the choice of dataset or data visualization graphs.
While I found the idea interesting, some of the findings/takeaways listed have not been exaggerated. For example, the study did not evaluate dependency, but it was mentioned that participants might get overly dependent on landmarks. This claim requires a long-term study. I suggest revisiting these statements.
The goal of study 2 is stated as understand the effect of landmarks in “more-complex” data visualization. This goal is vague; more complex than what? How was the “complexity” defined?
The second limitation is framed as future work, not a limitation of the study.
The paper is long with a relatively small contribution. Many repeated parts could be removed; I suggest shortening and removing the repeated sections.
Overall, the paper suggests 3 contributions that could be beneficial for future visualization designs. Upon applying the changes requested, I recommend accepting the paper.

Small notes:
In the introduction, it was stated there are 4 contributions, but only 3 were listed.
It would be great to add the reference in this sentence” Although previous research suggests that targets such as NAM-18 and NAM-2 should have been much more difficult in the No-Landmarks condition (where there were no visual features to help users remember these locations), search times actually favoured the No-Landmarks condition.”
Similarly, here “By contrast (and as expected from previous literature on learning), search performance continued to improve in the No-Landmarks condition:” the reference to literature is missing.
Fig 12 is stretched.

---

### Official Review · Reviewer_dWkY · 2022-04-14
**Good insight**

**Rating:** 8
**Confidence:** 4

**Review:**

The accidental landmark problem is justified with related work and is also logical--once stated. Yet, it is something I've never considered in decades of working on NPR and various navigation UIs! So, this is a great topic.

The studies reported make a careful and convincing case in support of the paper's hypothesis. They do not "solve" the problem, but the point of the paper is to make a scientific description of a relatively novel problem in this space for future work to then incrementally resolve.

I found the perceptual and infovis summaries helpful and appropriate despite being relatively long for a human interaction style paper and applaud the authors for the detail.

The sample sizes and diversity of subjects is appropriate for the work and gives the results credibility. Crowdsourcing of course provides some bias and limits control over the experimentation environment, but as the paper itself points out, has become a standard and especially during COVID has been a good way to continue to advance human studies without risk to the subjects or experimenters.

---

### Decision · Program_Chairs · 2022-04-17

Accept